# Technological Advancement for Efficiency Enhancement of Biodiesel and Residual Glycerol Refining: A Mini Review

**Nurhani Fatihah Jariah [1], Mohd Ali Hassan [1], Yun Hin Taufiq-Yap [2,3] and Ahmad Muhaimin Roslan [1,\*]**

1. Department of Bioprocess Technology, Faculty of Biotechnology and Biomolecular Sciences, Universiti Putra Malaysia (UPM), Serdang 43400, Selangor, Malaysia; nurhanifatihah@gmail.com (N.F.J.); alihas@upm.edu.my (M.A.H.)
2. Catalysis Science and Technology Research Centre (PutraCAT), Faculty of Science, Universiti Putra Malaysia (UPM), Serdang 43400, Selangor, Malaysia; taufiq@upm.edu.my
3. Faculty of Science and Natural Resources, Universiti Malaysia Sabah, Jalan UMS, Kota Kinabalu 88400, Sabah, Malaysia
* Correspondence: ar_muhaimin@upm.edu.my; Tel.: +60-3-9769-8054

**Abstract:** Biodiesel or known as fatty acid methyl ester (FAME), is a diesel fuel substitute derived from the transesterification reaction of triglycerides with alcohol in the presence of suitable catalyst. The demand for biodiesel is increasing due to environmental and health awareness, as well as diminishing energy security. However, the presence of impurities in biodiesel will affect engine performance by corroding fuel tubes and damaging the injectors. Common methods for the purification of biodiesel include water washing, dry washing and membrane separation. This mini review compares the technological advancement for efficient enhancement of biodiesel and glycerol refining between wet washing, dry washing (activated compound, biomass-based adsorbents and silica-based adsorbents), ion exchange and membrane separation technology. The percentage of glycerol residues, soap, alcohol and catalyst from crude biodiesel was compared to reflect the resulting biodiesel purity variation. The advantages and disadvantages of each method were also discussed.

**Keywords:** biodiesel; glycerol; purification; wet washing; dry washing

## 1. Introduction

Nowadays, there is an increasing trend in global energy demand due to rapid industrialization and population growth [1]. By 2050, the world's population is projected to reach approximately 9.9 billion [2]. Naturally, the energy consumption will increase with the growing population. Based on a statistical review of world energy in 2020, the estimation of total world biodiesel production is 669 thousand barrels of oil equivalent per day [3]. Extensive utilization of the fossil fuel-based is responsible for the emission of greenhouse gases especially carbon dioxide ($CO_2$) which contribute to the global warming and climate change. Therefore, tremendous efforts are currently being made to find the best alternative fuels that could reduce environmental deterioration and energy consumption. Renewable sources such as biofuels (biodiesel, biogas and bioethanol) are adopted as substitute for conventional petroleum fuel [4,5]. Since biofuels are carbon neutral, they can reduce the utilization of fossil fuels and subsequently, reduce the emission of $CO_2$. In fact, when biofuel is burned, the $CO_2$ is released to the environment and being used by plants for photosynthesis [6].

Furthermore, biofuels could be an excellent alternative because waste feedstocks are readily available for their production through various chemical and biological techniques. Among the biofuels, biodiesel has attracted extensive research worldwide [7]. Biodiesel, also known as fatty acid methyl ester (FAME), is a diesel fuel substitute produced by transesterifying triglycerides, which are the main component in many feedstock oils such as edible and non-edible oils, with alcohol in the presence of a catalyst [8]. Normally,

different fatty acids can be found in the triglycerides of vegetable oils and animal fats. The properties of the oils are influenced by the physical and chemical composition of the fatty acids. Figure 1 shows the chemical reaction involved in the production of biodiesel. Biodiesel has numerous advantages over petroleum diesel such as green, sustainability, biodegradability and clean-burning fuel [9,10].

$$
\begin{array}{c}
CH_2 - O - COR_1 \\
CH - O - COR_2 \\
CH_2 - O - COR_3
\end{array}
\; + \; 3\,CH_3OH \longrightarrow
\begin{array}{c}
CH_2 - OH \\
CH - OH \\
CH_2 - OH
\end{array}
\; + \;
\begin{array}{c}
CH_3 - O - COR_1 \\
CH_3 - O - COR_2 \\
CH_3 - O - COR_3
\end{array}
$$

Triglyceride          Methanol          Glycerol          Biodiesel

**Figure 1.** Reaction of biodiesel formation.

The use of low-grade oil as the feedstock could reduce the fuel quality. Therefore, further purification of crude biodiesel is needed to remove the impurities such as remaining vegetable oil, alcohol, catalyst, soap and free fatty acids to satisfy the standard specifications prescribed in the European Standard (EN), EN 14214 and the American Society for Testing and Materials (ASTM), ASTM D6751 [11]. According to Demirbas [12], impurities will result in low-quality biodiesel, thus affecting the engine performance and leading to storage and transportation problems. Therefore, this mini review focuses on the various biodiesel purification techniques and methods.

## 2. Conventional Protocols

### 2.1. Wet/Water Washing

Soap, catalyst, methanol and other pollutants are removed during conventional biodiesel refining. Since both methanol and glycerol are highly soluble in water, one of the most popular biodiesel purification methods is wet / water washing. It can eliminate any leftover compounds and soap generated in the biodiesel production effectively [13]. However, water washing has many disadvantages. This technique utilizes a lot of water, thus resulting in large amount of effluent. Furthermore, free fatty acids and stable emulsion will be formed. This technique also requires a drying procedure which is energy and time-consuming [14,15].

A research conducted by Chongkhong et al. [16] highlighted that the purification step is needed because the transesterified product obtained still had 1.4 wt% of free fatty acid. In that research, the refining experiment utilized the neutralization technique rather than the distillation technique to purify the FAME product. To discard soap formation, the procedure began with 3 M sodium hydroxide in water, followed by dissolving the solution with 2 wt% sodium chloride. The ester phase was washed with 60–80 °C water and allowed to settle before heated to evaporate the remaining water.

Another study by Yang et al. [17] proposed that biodiesel produced from camelina should undergo a refining process using distilled water. Temperatures of 20 °C and 50 °C were set for the distilled water; however, the effect of water temperature on the purification efficiency was insignificant. The water washing technique works based on the removal of water-soluble residues. For example, warm distilled or deionized water has a potential to discard residual glycerol, soap, alcohol and catalyst from crude biodiesel.

Another study reported that biodiesel production can be performed using simple filtration of cation-exchange resin catalyst [18]. The resulting methyl ester was washed in hot deionized water and dried with anhydrous sodium sulfate ($Na_2SO_4$). The study also suggested that tetrahydrofuran can also be used as co-solvent in the transesterification process since it has the capability to improve the homogeneity of oil and alcohol. Nevertheless, more processing equipment is required to remove the co-solvent. Thus, by performing water washing including acid neutralization, sodium hydroxide and other residues could

be eliminated. This includes excess alcohol, triglyceride, diglyceride and monoglyceride. However, it is noteworthy that the downstream process especially glycerol separation and refining step could be complex and costly [19].

In another work, three different types of water were used to purify biodiesel namely deionized water, tap water and acidified water (5% phosphoric acid) [13]. It was reported that the water washing technique could reduce both free glycerol and methanol even surpassing the biodiesel standard EN14214. However, this technique did not give any effect on glyceride removal. Nevertheless, at any temperature either ambient or 60 °C, it is possible to remove methanol. This is because as the washing temperature increases, the diffusivity of glycerol from biodiesel to water phase also increases, thus resulting in higher mass transfer coefficient.

### 2.2. Physicochemical Treatment

The most popular conventional method for purifying crude glycerol involves three steps: neutralization to extract soaps and salts, vacuum evaporation to remove methanol and water and final purification to improve glycerol purity [20]. Usually, neutralization process is used in the pre-treatment for the crude glycerol purification. In this process, a strong acid is used to remove the catalyst and soaps. Free fatty acid will be produced by reacting acid with soap, while reacting free fatty acids with a base catalyst produces salt and water. Some salts and insoluble free fatty acids will float on top and are easy to skim off. Table 1 shows the comparison between physicochemical treatments of crude glycerol.

Purification method of the crude glycerol by combining both physical and chemical treatments with solvent extraction has also been reported [21]. The benefits of the method are the used solvent can be reused, high efficiency with low production cost and easy to handle. That study also reported that the repeating acidification (pH 1–6) process for crude glycerol from used-oil waste increased the amount of glycerol rich layer while decreasing the amount of inorganic salt and free fatty acids. After conducting chemical process followed by neutralization with 12.5 M NaOH and physical treatment, the purity of the glycerol obtained was approximately 93.94%, and that the pH and matter organic non-glycerol (MONG) were also lowered (5.15% *w/w*).

Ooi et al. [22] used glycerol residue waste to recover crude glycerine, crude fatty acids and salts. It was reported that the crude glycerol was successfully recovered using 6% (*v/v*) $H_2SO_4$. The results obtained is in line with Kongjao et al. [21] in which chemical treatment at low pH yielded higher glycerol recovery. However, the ash content was reduced while there was slight increase in the MONG and water contents. This might be due to the insufficient drying of the glycerine. MONG consists of FAME, tri, di and monoglycerides, various forms of fatty acids and methanol. This treatment can also recover a large amount of salt while lowering crude glycerine levels, and at the same time leaving crude fatty acid recovery unaffected. Tianfeng et al. [23] suggested that the acidification of crude glycerol from transesterification of used cooking oil will increase the volume of glycerol-rich layer from 40% to 70%. They used 5.85% of phosphoric acid ($H_3PO_4$) at pH 5–6. Following acidification for one hour at 70 °C, the product yield of glycerol was 81.20% purity and the subsequent treatment by using vacuum distillation yielded glycerol purity of 98.10%.

In glycerol acidification process, the pH of crude glycerol was adjusted using strong acids (phosphoric, sulfuric, hydrochloric and acetic) to convert soap into fatty acid and salts [24]. It was found that when using the neutralization process, the purity of the glycerol was approximately 86 wt%. On the other hand, when using both saponification and neutralization, the glycerol purity was improved up to 99 wt% (only free fatty acids). Javani et al. [25] suggested that the potassium phosphate should be produced from the series of reaction of saponification and repeated acidification of crude glycerol. They proposed the method to reduce the biodiesel production while generating high quality of purified glycerol and free fatty acids since the price of the product is expensive.

**Table 1.** Comparison between physicochemical treatments of crude glycerol.

| Source of Crude Glycerol | Purification Method | Result | References |
|---|---|---|---|
| Used-oil waste | • Acidification with 1.19 M $H_2SO_4$<br>• Neutralization with 12.5 M NaOH<br>• Physical treatment | • 69% of glycerol removal with 93.94% purity<br>• 90% MONG removal<br>• Water content reduced from 6.7 to 1.5% (*w/w*) | [21] |
| Glycerol residue from palm kernel oil methyl ester plant | • Acidification with 6% $H_2SO_4$<br>• Neutralization with 50% NaOH<br>• Physical treatment | • 66% of glycerol removal<br>• 32% MONG removal<br>• Water content increased from 5.9 to 8.9% (*w/w*) | [22] |
| Transesterification of used cooking oil | • Acidification with 85% $H_3PO_4$<br>• Vacuum distillation | • 98.10% (*w/w*) glycerol purity was achieved following decolorization using 2% of active carbon | [23] |
| Raw glycerol phase | • Acidification with strong acids (phosphoric, sulfuric, hydrochloric and acetic)<br>• Neutralization with KOH<br>• Gravitational separation | • Glycerol with a purity of 86 wt% was achieved via neutralization<br>• Glycerol with a purity of 99 wt% was achieved via saponification plus neutralization | [24] |
| Transesterification of waste cooking oil | • Acidification with phosphoric acid<br>• Neutralization with 10 M NaOH<br>• Physical treatment | • 58% of glycerol removal with 96.08% purity<br>• 99% MONG removal<br>• Water content reduced from 14 to 0.03% (*w/w*) | [25] |

*2.3. Distillation*

Distillation is one of the glycerol purification methods. It is a process of purifying two or more relatively volatile liquids by process of heating a liquid to boiling, and then collecting their vapors to condense them in liquid state. Distillation is ineffective for feed streams that are sensitive to thermal degradation or polymerize at high temperature. Therefore, vacuum distillation is an option to solve the problem by reducing the pressure in the column, thus lowering the boiling point of the liquid [26].

Glycerol polymerization into polyglycerol occurs at temperatures above 200 °C, while dehydration occurs at temperatures above 160 °C in slightly acidic conditions, and glycerol oxidation into glycerose, glyceraldehydes and di-dihydroxyacetone often occurs [27]. Therefore, the purification step should be conducted under vacuum condition to prevent the degeneration of glycerol. Under vacuum condition, pH, temperature and pressure are being controlled. Vacuum distillation is widely used to purify glycerol.

Based on the study by Yong et al. [28], crude glycerol was purified using vacuum distillation. The crude glycerol was purified or distilled at lower temperature and pressure in order to avoid the unwanted reactions as previously mentioned. From this process, the purified glycerol was analyzed for several parameters such as ash, water and MONG which yielded 0.03% ash, 1% water and 2.4% MONG. In addition, it was identified that the ideal pH of refining was less than 5, which prevented foaming. The vacuum distillation performed successfully purified the crude glycerol up to 96.6% purity.

Distillation is a well-known technology for industrial glycerol refining because it can be used for small to large-scale continuous operations, low chemical costs and process versatility to suit varying raw and final product qualities. Nevertheless, it should be noted that the crude glycerol distillation process has several drawbacks, for example, it requires high energy input and has high maintenance [29]. Since glycerol has a high specific heat

capacity, the energy is used for vaporization, thus resulting in thermal decomposition, which will increase the plant operating costs by about 50% [20].

## 3. State of the Art Technologies

### 3.1. Dry Washing by Adsorption

In the dry washing process, impurities or residues from the liquid mixture will attach onto the surface of the adsorbent. Dry washing has many benefits as compared to the wet washing such as the reduction of aqueous effluents, shorter purification time and environmentally friendly [30]. Adsorption can be defined as the process by which the adsorbates adhere to the adsorbent, as illustrated in Figure 2 [31]. Surface characteristics such as specific surface area and pore structure are the main factors affecting the physical adsorption process [32]. In addition, adsorption process only occurs on the active site of the adsorbents. In the biodiesel purification, the adsorption process can be divided based on the types of adsorbents. The most common adsorbents used for this process are biomass-derived adsorbents such as (ligno)cellulosic substrates, activated compound such as activated carbon (AC), activated fiber and activated clay, and silica-based adsorbents such as Magnesol and Trisyl [33,34].

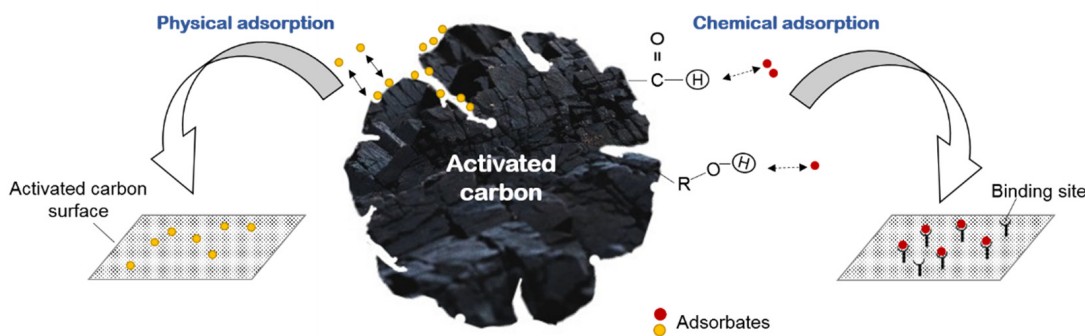

**Figure 2.** Interaction between activated carbon surface (adsorbent) and adsorbate molecules by physical and chemical adsorption.

#### 3.1.1. Activated Adsorbent Compound

Activated carbon is widely used to purify biodiesel. Any carbonaceous material such as oil palm biomass, bamboo, sawdust, lignite, coal, coconut shells, willow peat and wood can be used to make activated carbon. This bio-adsorbent has unique characteristics such as large porous volume and high surface area [35]. There are three factors affecting the adsorption process especially in biodiesel purification namely surface functional groups, pore size and surface area of the adsorbents. According to Fang et al. [36], the adsorption capacity is improved by increasing the basic functionalities of the adsorbents such as hydroxyl group. Basic functional groups are usually obtained by reducing oxygen functional groups or introducing nitrogen functional groups on the carbon surface. Furthermore, to ensure that the impurities in the biodiesel have accessibility towards the adsorption site, pore sizes of the adsorbent should be wide enough [37]. Alnaief et al. [38] compared the effect of different particle sizes ($\leq$90 µm and $\leq$150 µm) of activated carbon on biodiesel purification. The results showed that 2% of adsorbent (olive cake alkali residue) with particle size $\leq$90 µm was able to remove free glycerin after 30 min of residence time. Whereas for the particle size $\leq$150 µm, the free glycerin was completely removed after 40 min of residence time. Particle sizes of $\leq$90 µm have higher surface area as compared to particle sizes of $\leq$150 µm.

According to Chaudhari and Dhobale [39], when using activated carbon for purification process, higher yields of FAME and better fuel properties could be achieved. In that study, Karanja oil was transformed into biodiesel via base catalyzed transesterification in an Oscillatory Baffled Reactor. After that, Karanja seed cake activated carbon was used to

purify the biodiesel to meet the standards. The purification process was accomplished in a column with 10 cm height and filled with activated carbon. Similar findings were achieved where Karanja cake adsorbents yielded better result in terms of biodiesel yields and flow properties. In the purification process of biodiesel, Fadhil and Dheyab [40] studied the performances of activated carbon before or after treatment with acid (sulfuric acid or hydrochloric acid) synthesized from spent fish frying oil (SFFO) and spent cooking oil (SCO). The authors compared the yield between purified biodiesel from activated carbon and from conventional wet washing. The results showed that biodiesel purification process by activated carbon gave higher yield (91.50–93.75%) as compared to wet washing method (86–89%).

Biodiesel purification can also be performed using activated carbon processed from spent tea waste in a chromatography column [41]. The chromatography column had an internal diameter of 1 cm, and a height of 15 cm, and was packed with glass wool. In the column, 2 g of activated carbon were added. After that, at a flow rate of 15 drop/min, the crude biodiesels were allowed to pass through the adsorbent bed. The results obtained were compared with other purification methods such as using water washing and silica gel. The purified biodiesel from activated carbon had better fuel properties within the prescribed standard biodiesel specification (ASTM D6751) as well as higher yields. Apart from that, the spent activated carbon derived from tea waste could be regenerated and reused for the same purpose or be used in soil amendments as fertilizer. Putra et al. [42] stated that activated carbon can also be used as bioadsorbents in the pre-treatment process of used cooking oil. The results obtained were similar to the previous studies in which activated charcoal was able to reduce the water content (<0.1%) and free fatty acids (0.23%) in comparison with clay minerals.

### 3.1.2. Biomass-Based Adsorbent

Starch and cellulose can also be utilized in the purification of biodiesel as adsorbents [43]. These materials are inexpensive, abundant and have many benefits such as non-toxic, biocompatible and biodegradable. This technique is generally utilized in small-scale biodiesel plants where biomass that adsorb the residues is burned for heating after saturation and refilled with fresh biomass.

Cellulose and starch are natural adsorbents derived from various types of resources such as corn, potato, cassava and rice. Gomes et al. [43] stated that these natural adsorbents could be applied for biodiesel purification derived from sunflower oil as the feedstock. Corn and rice starches have a polyhedral structure, while potato and cassava starches have ellipsoidal and semi-spherical structures, respectively. In addition, for the purification process, eucalyptus bleached kraft cellulose with a tape format has also been used. The purification process is conducted by adding different amounts of adsorbents ranging from 1% to 10% in biodiesel for 10 min at 25 °C with agitation speed of about 250 rpm followed by filtration using filter paper. After purification with the aforementioned biomasses, the acidity index decreases. By using substrates derived from 1% rice starch, 1–2% cassava starch and 5% potato starch, the free glycerol content in the biodiesel can be reduced to 0.13%. Free glycerine in the purified biodiesel is also within the standard specification when using 2% corn starch. Turbidity of the biodiesel decreases when using 1–2% cassava starch or 5% potato starch as the adsorbents. Therefore, the utilization of natural adsorbents (cellulose and starch) for biodiesel refining has great potential to be applied in industrial scale since it has the ability to remove contaminants at 25 °C [43].

Rice husk is a waste product from grain processing, while rice husk ash (RHA) is a residue that is produced under controlled burning of rice husk [44]. RHA can also be used as alternative adsorbents in biodiesel purification. Manique et al. [45] used RHA at varied concentrations (1–5%) as adsorbents to purify the biodiesel synthesized from waste frying oil (WFO), and compared those with commercial adsorbents Magnesol® (1%) and conventional acidified water (1% aqueous $H_3PO_4$). The presence of meso- and micropores in RHA structure, as well as a high silica content, contribute to its high adsorption capacity.

From the results obtained, the water content in the purified biodiesel was lower than in that purified using 1% Magnesol® and 1% aqueous $H_3PO_4$.

In a dry purification process, sugarcane bagasse can also be used as a low-cost adsorbent for biodiesel refining [46]. The authors used three different forms of bagasse as adsorbents namely sugarcane bagasse ash, raw sugarcane bagasse and steam explosion pre-treated sugarcane bagasse. In addition, comparison between the purified FAME using other forms of sugarcane bagasse and a commercial solid adsorbent (Magnesol®) was also studied. About 3 wt% of the adsorbent's concentration were added to crude biodiesel, and the adsorption process was carried out under agitation speed of 120 rpm at 30 °C for two hours. The results obtained showed that the amount of glycerol content in biodiesel surpassed the standard limit of lower than 0.02 wt%.

Bioadsorbents derived from organic residues such as sawdust, coconut coir, nutshell, rice husk and water hyacinth fiber have also been used in biodiesel purification. In this study, these bioadsorbents purified biodiesel from waste cooking oil. From the result obtained, only bioadsorbents derived from water hyacinth fiber, rice husk, sawdust and coconut coir were able to reduce acid number to levels below the standard specification in ASTM D6751. It was demonstrated that some of the bioadsorbents had the capability to remove free fatty acids from biodiesel with high acid number. In terms of total glycerine, the value measured in the purified biodiesel was slightly higher than unpurified biodiesel. It might be due to the moisture content in the bioadsorbents. Nevertheless, these values surpassed the ASTM standard which is 0.24% mass. Among the bioadsorbents used, only sawdust showed a better result in biodiesel purification by removing contaminants such as glycerine, free fatty acids and water that satisfied the ASTM limits [38]. Table 2 shows the purification of biodiesel using various types of biomass-based adsorbents.

**Table 2.** Crude biodiesel refining using biomass-based adsorbents.

| Reactants Used in Biodiesel Production | Purification Method | Operating Condition | Finding | References |
|---|---|---|---|---|
| Sunflower oil, methanol, NaOH | Natural adsorbents (corn, potato, cassava and rice) | Sugarcane bagasse as adsorbents | Acidity index, combined alkalinity, free glycerine and turbidity decreased | [43] |
| Waste frying oil, methanol, KOH | Rice husk ash (RHA) as adsorbents | By varying the concentration of RHA adsorbents (1%, 2%, 3%, 4%, 5% *w/w*) at 65 °C for 20 min of stirring | At 4% rice husk ash, an efficient removal of glycerine and glycerides, potassium were achieved | [45] |
| Soybean oil | Sugarcane bagasse as adsorbents | Adsorbent loading ranged from 0.1 to 3 wt%, and stirred at 120 rpm at 30 °C for two hours | 3 wt% of sugarcane bagasse adsorbents were able to remove 82% of glycerine from crude biodiesel, thus yielding 87% | [46] |
| Waste cooking oil | Sawdust, coconut coir, nutshell, rice husk and water hyacinth fiber | 5% of bioadsorbents was stirred for 20 min at 700 rpm | 5% of sawdust decreased the acid number, water content and free glycerine content of biodiesel with values below the ASTM standard | [38] |

### 3.1.3. Silica-Based Adsorbent

Silica is another type of adsorbent used in dry biodiesel purification. Silica-based adsorbents have broad range of applications such as in sensor technology, catalyst support and gas storage [47]. Silica-based adsorbents have many properties such as high porosity, large internal surface area and high adsorption properties of the accessible materials. Silica gel, zeolites and molecular sieves are other types of industrially available adsorbents [32]. The polymerization of silicic acid will form an amorphous inorganic mesoporous adsorbent. A simple refining method was reported using silica to remove glycerides and free fatty

acid from the biodiesel product [48]. The biodiesel production was generated through non-catalytic supercritical methanol [49]. Therefore, among virgin activated carbon, impregnated activated carbons, diatomaceous earth and silica; the best adsorption capacity was showed by silica as it adsorbed high amount of FFA from the biodiesel, and subsequently, could be the best adsorbent choice in industrial bleacher. Silica gel gives both competent and robust characteristic in adsorbing glycerol and monoglycerides.

Other than that, silica hydrogel can aid in removing certain impurities in the biodiesel such as unreacted methanol, monoglycerides, diglycerides and moisture. These residues are produced from the transesterification process [50]. Silica hydrogel, also known as aqua-gel, is so named because it has high moisture content of about 60–65 wt%. Since silica hydrogel has high adsorption capacity and affinity towards phospholipids, trace metals and soaps, it is a good alternative in edible oil refining process. Yori et al. [51] reported that silica showed strong ability to eliminate glycerol from crude methyl ester by utilizing used cooking oil as feedstock. This was supported by similar research carried out by Predojević [52] in which the purification of crude methyl ester was accomplished by silica gel column chromatography. The biodiesel was passed through a column of 2 cm internal diameter and 2 cm height with 3 g of silica bed and 1 cm of anhydrous sodium sulphate at the top to remove any water traces. Results showed that the highest yields of biodiesel (~92%) was obtained using silica gel and phosphoric acid treatment as compared to hot water treatment (~89%). Table 3 summarizes the functions of different silica-based adsorbents involved in biodiesel refining process.

**Table 3.** Comparison of different silica-based adsorbents and their functions.

| Types of Silica-Based Adsorbent | Function | References |
|---|---|---|
| Silica | To remove glycerides and free fatty acid | [49] |
| Silica hydrogel | To remove unreacted methanol, monoglycerides, diglycerides and moisture | [50] |
| Fixed silica beds | To remove glycerol from a mixture of glycerol and purified biodiesel | [51] |

### 3.2. Dry Washing by Ion-Exchange

Ion exchange is known as the process of alternating the ions between the solution and a solid exchanger because of the stronger affinity between the target species and the functional groups on the surface. The ion exchangers comprise of a matrix with excess charges bound in specific sites of the structure [53]. Ion exchange resins are among the most widely used exchangers, and they are often made by functionalizing a polymer obtained by copolymerizing styrene cross-linked with divinylbenzene [54].

Ion exchange resins fall into two broad classifications which are anion exchangers and cation exchangers. These are further divided into strong basic and weak basic anion exchangers, and strong acidic and weak acidic cation exchangers. They are categorized based on their functionality and strength. In order to determine a good ion exchange resin, some consideration need to be taken such as the structural properties (e.g., degree of cross-linking, porosity and particle size), exchange capacity, stability, type and density of the charges (strength of the resin) [53].

Purolite® (PD206) is a commercial cation-exchange resin that is used to purify biodiesel by removing residual catalyst, water and other impurities, as well as acting as an adsorbent. A study conducted by Hazmi et al. [9] used PD206 and BD10 dry ion-exchange resins (strong acid cation resins) to refine biodiesel derived from used cooking oil and rapeseed oil. Results showed that the resins had better removal for soap and glycerol; however, methanol removal was low and did not pass the biodiesel standard EN14214. In contrast, Lewatit GF202 was able to remove methanol and potentially reduce glycerol content and soap in purified FAME. This resin can also be reused [55]. Lewatit GF202 column was also

used by Mata et al. [56] to purify biodiesel derived from beef tallow, pork lard and chicken fat. Results showed that the resin reduced biodiesel acidity and kinematic viscosity.

Kouzu and Hidaka [57] reported that biodiesel could be produced by the transesterification process of waste cooking oil with calcium oxide (CaO) as a catalyst. After the transesterification, the CaO catalyst in the biodiesel was removed by employing four resins namely Amberlyst-15DRY, Amberlite-200CTNA, Amberlite-IRC76 and Amberlyst-31WET. The procedure was performed by allowing the biodiesel containing leached CaO to pass through the column packed with cation-exchange resin. This procedure reduced free glycerol concentration to below 0.02 wt% to meet the EN14214 quality standard. This could be due to the fact that ion-exchange resin could easily absorb the polar organic liquids.

Examples of industrial ion-exchange resin are T45BD and T45BDMP (Thermax). Wall et al. [58] used both resins to study the biodiesel purification together with BD10 dry resin. In that study, filtration, physical adsorption, ion exchange and soap removal were the key factor. It was found that the resins gave the best performance in terms of soap removal as compared to potassium soap, and the process could be improved by decreasing the particle size of sodium soaps.

Dias et al. [59] used PD206 to purify biodiesel from soybean oil and waste frying oil. In that study, the effect of different resin contents (2% to 40%) on the ester and water traces present in the final FAME samples was investigated. At ambient temperature, the purification process was implemented for 60 min. The best quality product was achieved when using 40% resin to purify the samples; however, the content of ethyl ester was lower than biodiesel specification standard (EN14214). Furthermore, the same experiment was conducted by varying resin temperatures between 26 °C and 65 °C, while the resin content (2% Purolite) was kept constant. Results showed that at low temperature (26 °C) better water removal in the biodiesel was observed.

### 3.3. Membrane Filtration

Membrane technology also can be applied for biodiesel purification. Principle of the biodiesel purification by membrane separation process is depicted in Figure 3. Generally, membrane technology can be classified into organic membrane, inorganic membrane and hybrid membrane.

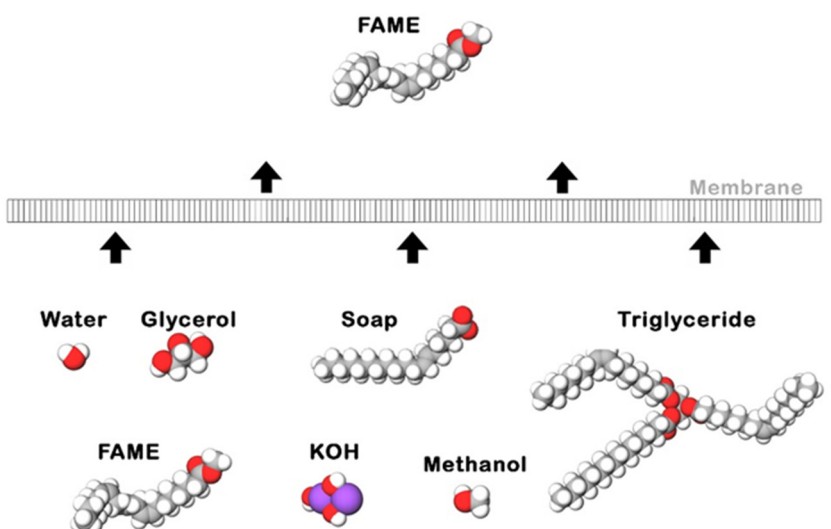

**Figure 3.** Biodiesel purification by membrane separation process.

### 3.3.1. Organic/Polymeric Membrane

Biodiesel can also be refined by using organic or polymeric membranes. The most common membranes used are polysulfone, polyamide, polycarbonate, regenerated celluloses, polyvinylidene fluoride and polyacrylonitrile with varying degrees of performance.

These organic membranes are classified based on their hydrophobicity or hydrophilicity. A hydrophilic membrane is more acceptable to alteration because of the fluctuation in pH and temperature; however, because of the water content, it is less susceptible to fouling from biorefinery feeds. For oil segregation, a hydrophobic material is more preferred [60].

He et al. [61] compared the purification of biodiesel using hollow fiber membranes (polysulfone and polyacrylonitrile), acid and water washing. All the three methods yielded the biodiesel purity of 97.5%; however, only the membrane extraction method could reduce the ester losses. Purification using distilled water at 50 °C resulted in 10.1 wt% loss of esters due to emulsification. Moreover, when comparing the ester losses between polysulfone and polyacrylonitrile membranes, the results showed 8.1 wt% and 10.3 wt% ester losses, respectively. In refining biodiesel, polysulfone membrane showed efficiency as compared to polyacrylonitrile membranes because it gave approximately 99% purity of biodiesel.

The application of ultrafiltration poly(ether-sulfone) and microfiltration cellulose ester membranes for biodiesel refining was conducted by Alves et al. [46]. Among the analyzed membranes, the ultrafiltration poly(ether-sulfone) membrane with a nominal molecular weight cutoff (NMWCO) of 10 kDa was able to obtain less than 0.02% of glycerol content in the biodiesel, thus surpassing the international biodiesel standard. Another study by Saleh et al. [62] found that by using ultrafiltration polyacrylonitrile membranes, 63% of the glycerol content in the biodiesel was successfully reduced due to the addition of 1 wt% of water by yielding 0.013 wt% glycerol in the biodiesel (permeate).

Biodiesel derived from microalgae cells also used organic membranes in the filtration step [63]. Algal cell separation also employed organic membrane and separation of protein and lipid from wastewater [64]. As studied by Giorno et al. [63], the fouling mechanism was tested using various organic membranes along with regenerated cellulose, polysulfone and polyvinylidene fluoride (PVDF). The tests were carried out using 40 mL of the microalgae *Nannochloropsis* sp. to test membranes between 100 and 150 kDa NMWCO for 20 min. The flux was then measured using water before and after contact between the membrane and the fresh algal cells. The result revealed that this organic membrane was highly favorable in the changing of pH and oil concentration [65]. At a similar kDa cutoff, the regenerated cellulose membrane was better than polysulfone membrane. Therefore, to enhance purification a smaller 30 kDa of regenerated cellulose membrane was selected. This 30 kDa regenerated cellulose membrane could reduce 89% of proteins as compared to using 100 kDa membrane, which could only remove protein to 61%. Higher fluxes and purities were achieved by further sonication of the algal cells.

### 3.3.2. Inorganic/Ceramic Membrane

Purification technology using ceramic membranes is rapidly gaining attention and interest due to their stability in organic solvents [66]. They are mostly preferred on an alpha-alumina support structure with titanium oxide or zirconium oxide [67].

Inorganic membranes offer several distinct advantages from organic membrane such as able to withstand harsh conditions such as high temperature and extreme pH, autoclavable, and have long lifetime. Due to these characteristics, inorganic membranes can react effectively with the base catalyst used in the transesterification process [34]. Table 4 summarizes the advantages and disadvantages of organic and inorganic membranes.

Saleh et al. [68] used 0.05 µm ceramic membrane at 25 °C to purify biodiesel and found that the glycerol content was less than 0.02 wt%, which met the standard EN14214. It seems that temperature is directly proportional to both permeate flux and separation. For example, at temperature up to 40 °C, the glycerol retention is increased to 99.2%. Furthermore, according to Wang et al. [69], the content of potassium and magnesium in the permeate flow can be lowered by reducing the pore size from 0.6 to 0.1 µm, which resulted in a 60% and 40% reduction in potassium and magnesium, respectively.

**Table 4.** Comparison between organic and inorganic membranes.

|  | Organic | Inorganic | References |
|---|---|---|---|
| Advantage | Cheap<br>Easy processing<br>Requires low energy in operation | Autoclavable<br>Long lifetime<br>Withstand high temperature (>200 °C)<br>Inertness to microbiological degradation<br>pH fluctuation resistance | [34,70,71] |
| Disadvantage | Short lifetime<br>Structurally weak, unstable,<br>temperature constrained | Fragile<br>Rigid<br>High capital cost | [70,71] |

### 3.3.3. Hybrid Membrane

The usage of hybrid membrane is one of the innovative technologies to improve the membrane performance. The membrane structure is created by a mixture of organic and inorganic materials. The inorganic component improves the mechanical and thermal properties, while the organic component retains the membrane flexibility. For instance, higher water permeability was achieved using a poly (methyl methacrylate) (PMMA)-$SiO_2$ composite membrane in which silica as tetraethoxysilane was added to the membrane formulation. The hybrid membrane would combine to improve water-soluble permeation, thus allowing for the most efficient purification of crude glycerine while maintaining the best mechanical and thermal properties. This resulted in much higher efficiency and sustainability [20].

One of the main issues in purifying biodiesel using membranes are low permeability and easy fouling. To overcome these, polyethersulfone-nano zinc oxide (PES-nano ZnO) nanohybrid membrane was developed [72]. The nanohybrid membrane was modified based on the composition of polyethersulfone (PES) and ZnO with exposure under UV irradiation. Results showed that the modified membrane could reduce the glycerol content from 4.00 to 0.36 wt%. This hybrid membrane had anti-fouling properties, better stability and increased flux permeability which could aid in refining biodiesel.

### 3.4. Comparison of Processes

The advantages and disadvantages of the water washing, dry washing by adsorption and ion-exchange processes and membrane filtration are summarized in Table 5.

**Table 5.** Comparison of different methods in biodiesel purification.

| Method | Advantage | Disadvantage | References |
|---|---|---|---|
| Water washing | • Most frequently used method | • Produces large amount of wastewater<br>• Extends production duration<br>• Increases production cost<br>• Reduces biodiesel quality by forming foam | [73,74] |
| Dry washing by adsorption and ion exchange process | • Waterless<br>• Strong affinity to polar compound<br>• Reduces purification duration<br>• Improves fuel quality | • Creates environmental and disposal problem since adsorbent such as magnesol and silica gel are non-recyclable.<br>• Ion exchange resin unable to remove glycerol and methanol | [73,75] |
| Membrane filtration | • Environmentally friendly<br>• Requires less energy<br>• Requires no chemicals<br>• Reduces number of processing steps<br>• Produces high quality final product | • Low permeability<br>• Easy fouling and clogging<br>• Undeveloped scale-up strategy | [72,74] |

## 4. Conclusions

The crude biodiesel can be purified by one of the following methods: wet washing, dry washing (activated compound, biomass-based adsorbents, silica-based adsorbents), ion exchange or membrane separation technology. Among these methods, the quality of the fatty acid methyl ester is the determining factor in identifying the most promising method. Wet washing is efficient in removing methanol and glycerol; however, large amount of wastewater will be produced. Due to the issue of wet washing, dry washing was established. Even though it is more environmentally friendly, there is concern in term of the disposal of spent adsorbents into the environment. Membrane separation technology is the best candidate for future development of biodiesel purification. Nevertheless, extra efforts are needed to manage the membrane fouling and clogging during the refining of biodiesel.

**Author Contributions:** Conceptualization, A.M.R. and N.F.J.; methodology, A.M.R. and N.F.J.; software, N.F.J.; validation, A.M.R., M.A.H. and Y.H.T.-Y.; formal analysis, N.F.J.; investigation, N.F.J.; resources, N.F.J.; data curation, N.F.J. and A.M.R.; writing—original draft preparation, N.F.J.; writing—review and editing, N.F.J. and A.M.R.; visualization, N.F.J.; supervision, A.M.R., M.A.H. and Y.H.T.-Y.; project administration, A.M.R.; funding acquisition, A.M.R. All authors have read and agreed to the published version of the manuscript.

**Funding:** This mini review received funding from GP-IPM/2016/9510900 and Graduate Research Fund (GRF). APC was funded by Research Management Center (RMC), Universiti Putra Malaysia (UPM), Malaysia.

**Institutional Review Board Statement:** Not applicable.

**Informed Consent Statement:** Not applicable.

**Data Availability Statement:** Not applicable.

**Acknowledgments:** The authors express their gratitude to Universiti Putra Malaysia for the technical and financial support by providing Graduate Research Fund (GRF) in the completion of this mini review.

**Conflicts of Interest:** The authors declare no conflict of interest.

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
