# Peer review of "Technological Advancement for Efficiency Enhancement of Biodiesel and Residual Glycerol Refining: A Mini Review"

_processes, doi:10.3390/pr9071198_

Round 1
Reviewer 1 Report
Review of “Technological advancement for efficiency enhancement of bio-2 diesel and residual glycerol refining: A mini review” submitted to Processes
The review needs major revison. The authors provided very detailed experimental procedures applied in other references without commenting on the general advancements that were the result of these works. This is supposed to be a review paper, not a collection of experimental parts from other documents. The experimental details are not necessary here. The review is lacking the comparisons between diferent works and the description of the actual advancements. The paper whould be rewritten in the form of a review, i.e. gathering the most crucial information from varoius papers, organising and comparing the information and remarking on the main advances. Experimental details can be always found in the original works; there is no need to copy it into the review, unless it is necessary for the conclusions/comparison of different approaches. Also, it is not clear what is the novelty of this review as compared to a wide range of works already published on this subject. The style lacks clarity and English needs revision. The manuscript contains plenty of errors typical of the young researchers/students, and it should be revised by a senior researcher, e.g. [24] suggested that …
1.Line 13-Biodiesel or known as fatty acid methyl ester (FAME) is a diesel fuel substitute derived from the transesterification reaction of triglycerides with alcohol in the presence of a relevant catalyst . FAME is a main component in many feedstock oils such as edible or non-edible oil.
In the abstract, the sentences are too long and difficult to understand. Try to divide the long phrases into two sentences for increased clarity.
- I disagree that the main interest in biodiesel is due to the depletion of fossil fuels. What about the environmental issues connected with the over-dependence on fossil fuels?
- line 21, substitute such as with by corroding and damaging the…
- Review the parts of the manuscript explaining why the energy market is turning to renewable carbon resources. The main driving forces to research renewables are: dreadful environmental impact of fossil fuels on the environment and the growing population around the world, and changes in lifestyles in developing countries leading to increased consumption of energy and products. The depletion of fossil fuels and rising prices of oil are not the primary concerns at the moment.
- line 44-can be found instead of “founded”
- line 48, biodegradability
- Line 47 Compared to petroleum diesel, biodiesel possesses many advantages such as acceptable properties for diesel engines? This sentence does not make sense. Petroleum diesel also has sufficient properties for diesel engine!
- figure 1-reaction of biodiesel formation
- Distillation is the process of purifying a liquid by process of heating and cooling? Of what?
- Line 146-this problem can be encountered-revise the sentance.
- Line 56 According to (name of the first author or corresponding author of the reference) (ref number), also in line 70, research conducted by (name of the author/s). Check the whole manuscript and add the name of the authors before the reference number when the senstance starts with as shown by, according to, another study by , a method discovered by, a study conducted by, as studied by… etc.
- In the introduction, the authors should mention other reviews of the same subject that are already published (e.g “A comprehensive review on biodiesel purification and upgrading” by Hamed Bateni, Alireza Saraeian and Chad Able) expaining what is the difference between what has been already reviewed and their mini review. The need for publishing such a review on the subject that has been studied on several occasions should be also explained.
- Line 77-82, what is the reason for copying experiemental procedure from another work? Some conclusions/main achievements should be discussed in the review paper, not the experimental methods from random works. In this case, which temperature was better and why it was better? Was the influence of the temperature confirmed by other studies? Line 93-100 the aforesaid. Remove the experimental details where they are not necessary.
- Line 93, revise the sentance.
- Line 133 conducted by…to perform –revise the sentence
- “These are the reasons why vacuum distillation is the most com-154 mon method for glycerol purification. In the purification of glycerol, vacuum distillation 155 is the most widely used process.”-revise the paragraph, avoid repetitions.
- Line 158-164-too many experimental details that are not needed. Revise the whole manuscript.
18.Line 168- it requires
- The description of how to activate char is not a subject of this review and should be removed. Instead, different carbon adsorbents should be described, containing varied surface chemistry. How different functional groups on carbon affect the adsorption process during purification? The authors should conclude the influence of the surface of these materials on the purification process, which is the subject of the review. There are many brilliant reviews dealing with the preparation of activated carbons.
20.Line 214 –which was
21-line 216 co-lumn
22-line 224- “Another literature studied by [40]”-revise how to reference the published works.
23-line 227-was able
24-line 232 such as they are
25-line 233 utilized
26-line 271 –processes were compared; what do the authors mean by simplified?
27-line 277-silica is another type of adsorbent
28-description of silica is not necessary; the authors should focus on the topic of the review.
29-Table 3. Compare organic with inorganic membranes; in two columns, compare the drawbacks and advantages of each of them. Swap Drawbacks and Advantages with Organic and Inorganic membranes
- More focus should be placed on reviewing the most modern technologies like hybrid membranes and biomass-derrived materials etc. Distillation and water washing are very well-known ancient technologies that should be only briefly mentioned in the review about advancements. There is no need to present long paragraphs about these methods. Out of 68 references in the review, only around 10 are from recent years (i.e.2020 or 2021). The authors should add the most recent papers (from the last few years) on the subject.
Author Response
We would like to thank the reviewers for their thoughtful comments and efforts towards improving the manuscript. In the attachment, we address the concerns of reviewers and make amendments as required. Thank you.

Reviewer 2 Report
Proposed article describing biodiesel and residual glycerol refining. This topic seems to be an actual point of interest. The manuscript is well organized, however, I found some issues, I listed below:
1) Lines 32-33 you stated that fossil fuels contribute to energy production decreasing because of their non-renewable nature. How about CO2 production?
2) Line 144 - distillation can be also done using pressure change, as you pointed later, therefore this statement should be corrected.
3) Lines 145 - 147 - you stated that thermal degradation can be encountered in a vacuum distillation? I think vacuum distillation is used to lower process temperature and solve this problem
4) Lines 153 -157 - this fragment should be rewritten. You wrote briefly: "vacuum distillation is common because it's widely used and this is the reason why is most popular" - I think no more comment is needed.
5) Line 173 - I suggest splitting "Dry washing" section into Dry washing by adsorption and dry washing by ion exchange because in the current version you stated, that dry washing can be performed using these two techniques, but under this section you only described adsorption process.
6) Lines 180-181 You should more precisely describe the adsorption (physical) process. You miss the key information about the process, especially that adsorption taking place inside porous in place that is called an active center or active site, not strictly anywhere at the surface. Moreover, Figure 2 is not appropriate and should be changed or removed.
7) Line 193 - Did you mean specific surface area?
8) Line 201 - Many of the citations you making of other works are wrong. Instead of "Based on [37]" or "[38] studied..." it should look like in line 236 "Gomes et al. [41] stated...".
9) Line 280 - What does it mean "internal surface area" and "high adsorption properties"? Specify which properties you are referring to.
10) In conclusion - could you state your own opinion - which method is better, or which trend is most promising and what should be done in future, to sum up, your review study?
Basing on the aforementioned comments I recommend the manuscript be reconsidered after major revision.
Author Response

(The authors gave the same response as above.)

Reviewer 3 Report
Please see the comments in the attached file.

Author Response

(The authors gave the same response as above.)

Reviewer 4 Report
This review reported the efficiency enhancement of biodiesel and residual glycerol refining. It reviewed the the purification method; water washing, dry washing (i.e., activated compound, biomass-based adsorbents and silica-based adsorbents), ion exchange, and membrane separation technology of biodiesel. Some comments for this review can be considered during the revision:
- Line 30, could you please update the biofule production data using the latest results after 2018?
- Line 54, how do you compare both standards?
- Line 75, please change "wt. percent" to "wt.%".
- Anybody figures showing the experimental results from the literature?
- In Figure 2, what are the differences between physical adsorption and chemical adsorption in terms of the adsorbate?
- In Section 3.3.3., what is status of the hybrid membranes? Any results presented using the membrane modules?
- In Section 3, the authors just simply introduced the different technologies. However, the drawbacks and challenges of these methods should be highlighted and discussed in detail. What technology will be the closest one approaching the large-scale application?
Author Response

(The authors gave the same response as above.)

Round 2
Reviewer 1 Report
The revised version is slightly better than the original manuscript. However, English continues to be very bad and needs significant improvements to reach the standards of scientific journals. Moreover, some of the conclusions in the article are common knowledge from school, the discussion of which should not be placed in a scientific journal. A senior researcher should appropriately revise the paper. It should have been done before the re-submission of the revised article.
Line 13- remove or and insert coma after (FAME)
Line 17-diminishing energy security
Line 21- the review compares (not presents) different methods of biodiesel purification
The abstract is still not satisfactory. In the abstract, the authors should underline the novelty of their work compared to other existing reviews focussed on similar subjects. In addition, English should be revised correctly.
Line 23- full stop is missing
Line 30- remove product
Line 32- “Thus, tremendous efforts have been made to find 32 the best alternative fuels that could reduce both energy and environmental degradation 33 by adopting biofuels generation from renewable sources”- what do the authors mean by reducing the degradation of the energy?
In the introduction, the authors should include the statistic about the steady growth of the population around the globe and increased need for energy and products. This is one of the primary reasons for turning into renewables…
Line 49- “Compared to petroleum diesel, biodiesel possesses many advantages such as biodegradability, clean-burning fuel, sustainable, green and renewable”-revise English.
Line 152- There is no need to put the basic definition of distillation. Distillation was studied in the secondary school and this is a scientific review, not the students’ textbook.
Line 170- of refining
Line 187-“The key element in the physical adsorption”- what do the authors mean by the key element? The main factors affecting the adsorption? Please revise the sentence.
Line 203- surface functional groups
Line 213- 30 minutes (changes mins for minutes in the whole manuscript
Line 212- was able
Line 212- “This can be deduced that, particle sizes of ≤ 90 μm have high surface area than ≤ 150 μm in size”. This sentence is not written in proper English. I have requested the authors to revise English in their manuscript, but obviously, this has not been done correctly.
It can be expected that smaller particles would have a higher specific surface area, so it does not have to be “deduced” .
Revise Figure 2. The mechanisms are not correct.
Table 4-membranes
Line 433- from 4.00 to 0.36 wt% with 91% reduction-revise the sentence putting either 91% reduction or 4% to 0.36%. It is not necessary to put both.
Line 472- purified by one of the following methods
Rewrite conclusions correcting style, and avoiding repetitions.
Author Response
We would like to thank the reviewers for their thoughtful comments and efforts towards improving the manuscript.
Thank you

Reviewer 2 Report
Thank you very much for the answers and corrections provided. I have no further comments.
Author Response

(The authors gave the same response as above.)

Reviewer 3 Report
My comments and questions were reflected and revised in the manuscript.
Now, the manuscript can be published.
Author Response

(The authors gave the same response as above.)
